# A single-cell atlas and lineage analysis of the adult *Drosophila* ovary

Katja Rust [1,2,3], Lauren E. Byrnes [1,4], Kevin Shengyang Yu[5], Jason S. Park[5], Julie B. Sneddon [1,3,4,6], Aaron D. Tward [5] & Todd G. Nystul [1,2,3]✉

The *Drosophila* ovary is a widely used model for germ cell and somatic tissue biology. Here we use single-cell RNA-sequencing (scRNA-seq) to build a comprehensive cell atlas of the adult *Drosophila* ovary that contains transcriptional profiles for every major cell type in the ovary, including the germline stem cells and their niche cells, follicle stem cells, and previously undescribed subpopulations of escort cells. In addition, we identify *Gal4* lines with specific expression patterns and perform lineage tracing of subpopulations of escort cells and follicle cells. We discover that a distinct subpopulation of escort cells is able to convert to follicle stem cells in response to starvation or upon genetic manipulation, including knockdown of *escargot*, or overactivation of mTor or Toll signalling.

[1] UCSF, Department of Anatomy, 513 Parnassus Ave, San Francisco, CA 94143, USA. [2] UCSF, Department of OB-GYN/RS, 513 Parnassus Ave, San Francisco, CA 94143, USA. [3] Broad Center of Regeneration Medicine and Stem Cell Research, 513 Parnassus Ave, San Francisco, CA 94143, USA. [4] UCSF, Department of Cell and Tissue Biology, 513 Parnassus Ave, San Francisco, CA 94143, USA. [5] UCSF, Department of Otolaryngology-Head and Neck Surgery, 513 Parnassus Ave, San Francisco, CA 94143, USA. [6] UCSF, Diabetes Center, 513 Parnassus Ave, San Francisco, CA 94143, USA. ✉email: todd.nystul@ucsf.edu

I n *Drosophila*, each ovary is composed of ~16 strands of developing follicles, called ovarioles, and oogenesis begins at the anterior of each ovariole in a structure called the germarium (Fig. 1a, b). Two to three germline stem cells (GSCs) reside at the anterior edge of the germarium in a niche produced by cap cells and terminal filament (TF) cells. GSCs divide during adulthood to self-renew and produce daughter cells called cystoblasts that move toward the posterior as they differentiate[1]. Escort cells (ECs, also referred to as inner germarial sheath cells) ensheath the cystoblasts and promote the early stages of differentiation as they undergo four rounds of incomplete mitosis to form into a cyst of 16 interconnected cells. One germ cell is

**Fig. 1 CellFindR identifies distinct populations of cells in the ovary. a** Diagram of the anterior tip of the ovariole, including the germarium and two budded follicles. **b** Diagram of the entire ovariole. **c** UMAP plot of the merged dataset and gene expression profiles of selected markers. **d** Hierarchy of CellFindR clusters. Tier 1 (dark brown outlines), Tier 2 (brown outlines), Tier 3 (taupe outlines) and Tier 4 (beige outlines) clusters were produced by the first, second, third and fourth round of CellFindR clustering, respectively. Orange branch lines indicate subclusters of a single terminal cluster that were identified through additional analysis. **e** Heat map showing the expression of the top 10 most unique genes for each cluster across the entire dataset. **f** Heat map showing the activity for selected regulons identified by SCENIC in each cluster. The first column shows regulon activity in dataset2 and the second column shows regulon activity in dataset3 for each regulon respectively. Scale bar shows percent of regulon activity. TF: terminal filament; GSC: germline stem cells; EC: escort cells; FSC: follicle stem cells; pFC: prefollicle cells; polar: polar cell; stalk: stalk cell; MB: main body follicle cells; St.: Stage; ant.: anterior; cent.: central; post.: posterior; undif.: undifferentiated.

selected to become the oocyte and enters meiosis, while the others differentiate into nurse cells that provide support for the oocyte. At the midpoint in the germarium, each cyst becomes encapsulated by a layer of epithelial follicle cells produced by the follicle stem cells (FSCs)[2]. The FSCs are known to reside at the anterior edge of the follicle epithelium, but the precise number and location of FSCs within the germarium has been debated recently. In particular, whereas our recent study supports the view that FSCs have low but detectable levels of the follicle cell marker, Fas3, and are thus on the posterior side of the Fas3 expression boundary[3,4], other studies suggest that FSCs are Fas3[−] and are thus anterior to the Fas3 expression boundary[5–7]. FSCs divide with asymmetric outcomes to self-renew and produce prefollicle cells (pFCs) that differentiate gradually, over the course of several divisions[8] into polar cells, stalk cells, or main body follicle cells (Fig. 1a). Newly budded follicles grow and develop into a mature egg over 4–5 days under ideal conditions[9,10]. This stereotypical process has been divided into 14 distinct stages[11], with early stages (Stages 1–6) characterized by rapid follicle growth and follicle cell division; mid-stages (Stages 7–10) characterized by the onset of yolk protein production, elongation of the follicle, growth of the oocyte, and specialization of follicle cells into subtypes such as stretch cells; and late stages (Stages 11–14) characterized by the death of nurse cells, deposition of the egg shell proteins, and growth of the oocyte to fill the entire volume inside the egg shell (Fig. 1b).

In this study, we use single-cell sequencing to build a comprehensive atlas containing transcriptomes and gene regulatory networks of all major ovarian cell types, including GSCs and FSCs and describe three subpopulations of ECs. We demonstrate the utility of the atlas by identifying cell type-specific markers and Gal4 driver lines. Using newly characterized EC drivers, we show that a subpopulation of ECs can convert to FSCs under severe starvation conditions or upon manipulation of *escargot* expression, mTor or Toll signaling.

## Results

**Transcriptomes and gene regulatory networks of ovarian cells.** To catalog the cell types in the *Drosophila* ovary, we performed scRNA-seq of ovaries from wildtype flies in triplicate (Supplementary Fig. 1a–c, Supplementary Table 1). This procedure produced transcriptional profiles of ~14,000 cells, achieving over 2× coverage of the ovariole (see "Methods"). We performed batch correction to merge the three datasets and clustered the cells using an adaptation of the Seurat algorithm[12,13] called Cell-FindR[14]. CellFindR performs the Seurat algorithm iteratively, first on the entire dataset, producing a set of "Tier 1" clusters, and then on each cluster separately to test whether further subclustering produces sufficiently distinct clusters to form a new tier on that branch. Since CellFindR produces sub-clusters independently for each cluster, this process achieves more reliable clusters than conventional clustering methods. Combining CellFindR with supervised sub-clustering produced 26 distinct clusters (Supplementary Tables 1–3) that can be arranged in a

hierarchical tree, with top-tier branches separating the most distantly related cell types and branches at each subsequent tier separating more and more closely related cell types (Fig. 1c, d). We found that this method was more accurate at producing clusters that aligned with markers of known cell types than using Seurat alone (Supplementary Table 2). Notably, the three datasets correlated well with each other ($r^2 > 0.96$) and all datasets contributed to nearly every cluster (Supplementary Fig. 1d–g, Supplementary Table 3), indicating that the methods were robust and reproducible. Using known and newly identified markers, we were able to assign the cell-type identity of all 26 clusters (Fig. 1c) and GO-term analysis further confirms cluster identities (Supplementary Data 1–2). We report distinct gene expression profiles for each of the 26 cell types (Fig. 1e, Supplementary Data 3–5). To identify regulons that are enriched in one or more clusters of cells in our dataset, we performed SCENIC analysis[15] on the two larger datasets 2 and 3 (Supplementary Data 6–7) and tested whether RNAi knockdown of transcription factors identified by the algorithm produced phenotypes in the ovary (Fig. 1f, Supplementary Fig. 2). This analysis identified many regulons that are active at specific stages of oogenesis, including some that are expected based on previous studies and others with previously undescribed roles in the ovary.

**Germ cell transcriptomes change rapidly during development.** The germ cells clustered apart from somatic cells into two terminal clusters on a single branch of the hierarchy tree that are distinguished by the expression of germ cell markers such as *vasa* (*vas*)[16,17] and the lack of expression of somatic cell markers such as *traffic jam* (*tj*) (Fig. 2a–c)[18]. One cluster is enriched for cells that express genes such as *bam* and *corolla* that are known to be expressed in germ cells within Regions 1 and 2a of the germarium, indicating that it corresponds to the earliest stages of germ cell development (Fig. 2d)[19–21]. The other cluster is enriched for expression of genes that become detectable in germ cells starting at Region 2b of the germarium, such as *oskar* (*osk*)[22], indicating that it contains germ cells at the next stage of differentiation (Fig. 2e). Germ cells at later stages of development are not included in our dataset because they are too big to be captured by our methods.

To estimate the lineage relationships among the germ cells in our dataset, we performed monocle3 analysis. Monocle3 is an algorithm that arranges cells along a bioinformatic trajectory that minimizes the differences in gene expression between neighboring cells[23–25]. When applied to a set of cells in the same lineage, the cells are organized in "pseudotime" according to the stage of differentiation (Fig. 2f, g, Supplementary Fig. 3a–c). Monocle3 arranged the germ cells into a linear trajectory that is consistent with the known progression of germ cell development, with cells expressing mitotic markers preceding those expressing meiotic markers (Fig. 2h). Moreover, it placed germ cells expressing genes involved in protein production at the latest stages, consistent with a role for nurse cells to produce cytoplasmic contents for the oocyte. The cells at the earliest stage of pseudotime have low

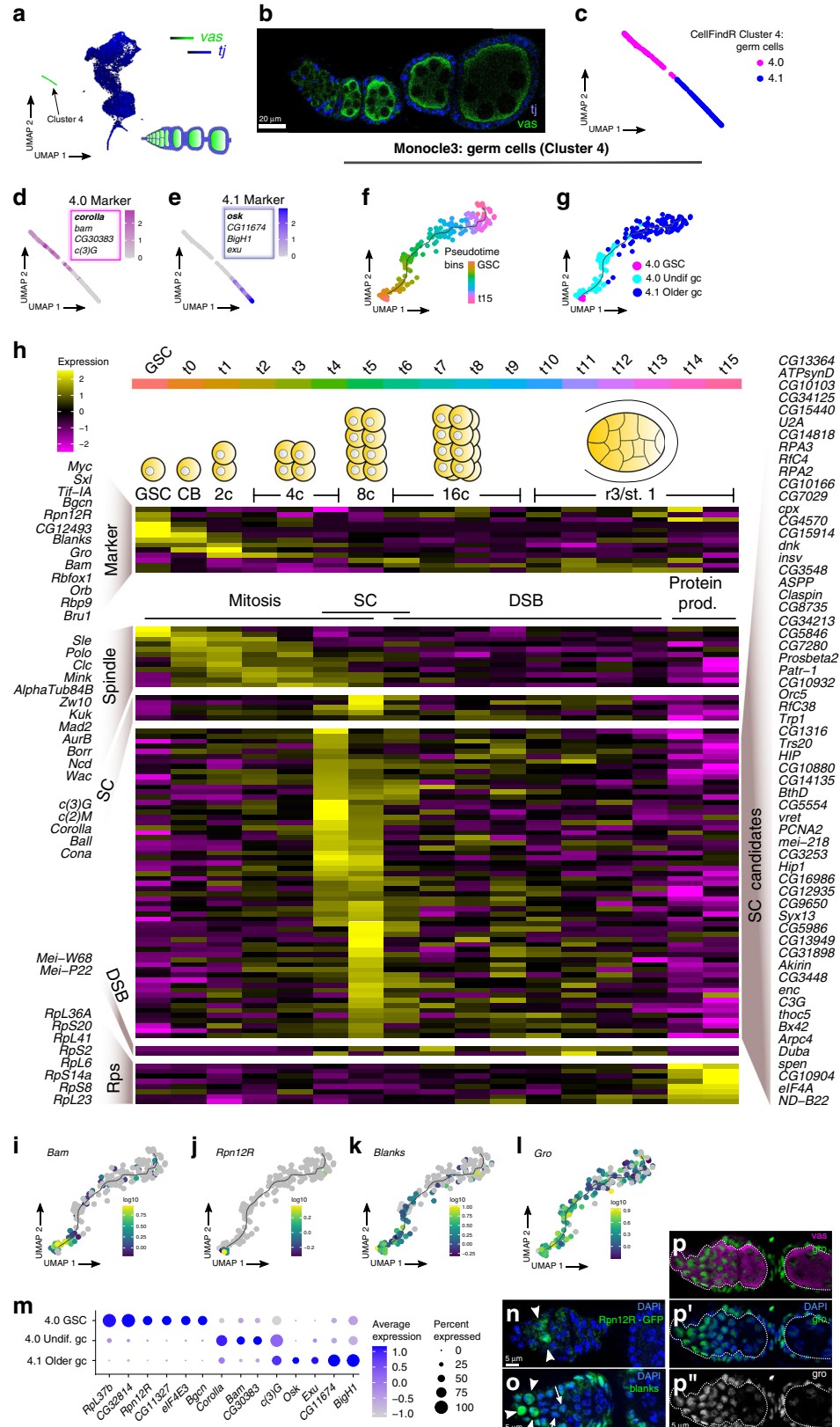

levels of the key cystoblast differentiation gene, *bag-of-marbles* (*bam*)[19,21], suggesting that they are GSCs (Fig. 2h, i). Consistent with this, we found that the top 100 most upregulated genes in GSC-like tumors[26] and many BMP response genes[27] are significantly enriched in germ cells at the earliest stage of pseudotime (Supplementary Fig. 3d–e). The markers of the

subsequent stages of germ cell pseudotime also align well with expectations from published studies. For example, the pseudotime analysis correctly predicted that *myc* is expressed in GSCs, downregulated in subsequent stages, and then upregulated again in 16-cell cysts[28], and that the onset of *orb* expression begins at approximately the 8-cell stage[29]. Likewise, *Tif-IA*, which

**Fig. 2 Germ cells. a** SCope expression plot of *vas* (green) and *tj* (blue) on UMAP plot and a diagram of an ovariole showing cell types in the corresponding colors. **b** Early stages of *Drosophila* ovariole stained for tj (blue) and vas (green). **c–e** UMAP plots showing the distribution of the two germ cell clusters initially identified by CellFindR (**c**), and the expression pattern of a marker for each cluster. Expression of the marker in bold text is shown on the plot and additional markers are listed below (**d–e**). **f–g** monocle3 analysis of germ cells orders cells into a linear trajectory (**f**) that distributes the cells from the two germ cell clusters onto opposite ends of the pseudotime trajectory and identifies GSCs (**g**). **h** Heat map showing transcriptional changes across pseudotime identifies markers of each stage of germ cell differentiation from the GSC to the Region 3/Stage 1 follicle, including stages that are enriched for the expression of mitosis genes, synaptonemal complex genes, double-stranded break genes, and protein production genes. Genes with a similar expression profile as known synaptonemal complex genes are presented as novel synaptonemal complex candidate genes. **i–l** The expression profile in pseudotime of representative markers of different stages of germ cell differentiation. **m** Dot plot showing the specificity of selected markers for GSCs, undifferentiated germ cells, and older germ cells. **n–p** Rpn12R-GFP germarium stained for GFP (**n**), or wildtype germaria stained for blanks (**o**) or gro (**p**, **p'**), shown in the green channel or in white (**p''**), and for DAPI (blue, **n**, **o**, **p'**) and vasa (magenta, **p**) as indicated. Arrowheads in **n** and **o** point at positive cells. Arrows in **o** point at germ cells with lower expression of blanks. White line in **p** demarks the border between germ cells and somatic cells. GSC: germline stem cell; undif.: undifferentiated; gc: germ cell; protein prod.: protein production; SC: synaptonemal complex; DSB: double-strand break; Rps: ribosomal proteins.

functions in GSC self-renewal[30,31], is predicted to be expressed in GSCs. We experimentally validated three candidate markers, *RpnL12R*, *blanks*, and *groucho* (*gro*) that are predicted to be expressed in GSCs and cystoblasts and then taper off at progressively later stages of germ cell development (Fig. 2j–m). Consistent with these expectations, we found that Rpn12R is detectable in the anterior-most germ cells within Region 1 of the germarium, blanks expression tapers off by the end of Region 1, and gro expression extends through Region 2a and into Region 2b (Fig. 2n–p). As previously reported, gro is also expressed in somatic cells[32], similar to the expression pattern of blanks. Collectively, this analysis allowed us to identify the transcriptional signature of each stage of germ cell development in the germarium and to identify dozens of previously undescribed candidate markers of germ cell development (Fig. 2h, Supplementary Data 8).

The remaining terminal clusters contain distinct somatic cell types, including all the major cell types of the ovary as well as hemocytes and muscle cells (Supplementary Fig. 4). One set of related terminal clusters contains all of the germarial somatic cell types, as well as polar cells and stalk cells. Interestingly, these clusters are enriched for GO terms involved in cellular morphogenesis and cytoskeletal dynamics, which may reflect the role of the germarium as the place where cellular rearrangements drive the formation of new structures (Supplementary Data 2). One terminal cluster is distinguished by the strong expression of the apical cell marker, *engrailed* (*en*)[33] (Fig. 3a–e). We identified two populations in this cluster that are distinguished from each other by several markers, including the cap cell marker *traffic jam*[18] (Fig. 3d, Supplementary Fig. 5a-b), indicating that the cluster contains both *en*+, *tj*+ cap cells and *en*+, *tj*− TF cells.

**Distinct EC subtypes defined by gradients of gene expression.** Three terminal clusters are distinguished by the expression of EC markers. While these three EC clusters shared many common EC genes such as *patched* (*ptc*) and *failed axons* (*fax*)[33–35] (Fig. 3c, d, Supplementary Data 9), they were clearly distinguishable by the expression of cluster-specific markers (Supplementary Fig. 5c–e), suggesting that they each contain a separate population of ECs. We used publicly available enhancer trap and protein trap lines to investigate the location of these EC populations. First, we confirmed that *ptc-GFP* and fax::GFP are expressed in all ECs, defined as the somatic cells in Regions 1 and 2a up to but not including the Fas3+ cells at the Region 2a/2b border[36,37], (Fig. 3f, g) and then assayed for markers that are differentially expressed in one or more clusters. ECs form cellular protrusions with a gradient of increasing protrusion lengths from the anterior to the posterior of the EC compartment[38,39]. We found that *Pdk1-Gal4* is strongly expressed in ECs throughout Region 1, including in ECs with

short and medium protrusion lengths, but is not detectable in the Region 2a ECs with long protrusions (Supplementary Fig. 5i-l); *hh-LacZ* is expressed in a decreasing gradient from the anterior to the posterior of the EC compartment[36,40]; *GstS1-LacZ* is expressed in ECs throughout Region 2a but not in Region 1; and *santa-maria-Gal4* is expressed weakly and sporadically in the ECs immediately adjacent to the Region 2a/2b border (Fig. 3h–j). We also found that castor (cas), which is strongly expressed in the early FSC lineage[41], is detectable at low levels in the ECs that are immediately adjacent to the Region 2a/2b border (Fig. 3g). As an additional test for distinctions in the transcriptional profiles of ECs, we performed monocle3 analysis on the entire EC population, and found that it identified three distinct EC populations that closely correspond with the three EC clusters identified by Cell-FindR (Supplementary Fig. 5f–h). Together, these observations indicate that there is an anterior-to-posterior gradient of EC identities that can be categorized into at least three populations: anterior ECs (aECs) that are Pdk1+; central ECs (cECs) that are GstS1+, cas−; and posterior ECs (pECs) that are GstS1+, cas+ (Fig. 3k, l).

To characterize these EC populations, we first determined the average size of each population per germarium. We found that an average of 39.8 ± 3.6 cells per germarium express the pan-EC marker *PZ1444-Gal4*, consistent with previous results[34], and that there are an average of 24.5 ± 3.3 *Pdk1-Gal4*+ aECs and 12.9 ± 2.1 *GstS1-LacZ*+ cECs and pECs. In addition, we found that there are an average of 2.5 ± 1.5 cas+ pECs, implying that there are ~10 GstS1+, cas− cECs (Fig. 3m). Next, to test the lineage potential of these EC populations, we combined EC specific Gal4 drivers with the lineage tracing tool, G-TRACE[42], in which RFP expression specifically labels the Gal4-expressing cells and GFP+ clones trace the lineage of the Gal4-expressing cells. To ensure that the G-TRACE tool is only activated during adulthood, we crossed in a *tub-Gal80ts* construct, raised flies at 18 °C, and shifted to 29 °C after eclosion (referred to herein as G-TRACEts). With *fax-Gal4*, which is expressed weakly in all EC populations, we observed an average of 4.4 RFP+ cells per germarium at 14 days post temperature shift (dpts). These cells were located in sporadic positions throughout the EC compartment, but never in Fas3+ cells, indicating that *fax-Gal4* is expressed in ECs but not in the FSC lineage. Likewise, we found that *fax-Gal4* driving G-TRACEts produced GFP+ ECs at 7 or 14 dpts, but did not produce GFP+ cells in the FSC lineage in nearly every case (Fig. 3n, Supplementary Fig. 5m). The only exceptions to this pattern were in four ovarioles that were isolated from the same fly. This interesting outlier is considered further below.

With *Pdk1-Gal4*, a driver specifically active in aECs, we observed strong RFP expression in Region 1 ECs but almost never in cECs or pECs in Region 2a (0.08%, $n = 1283$ RFP+ cells, 7dpts), as expected, and found that the GFP+ EC clones were

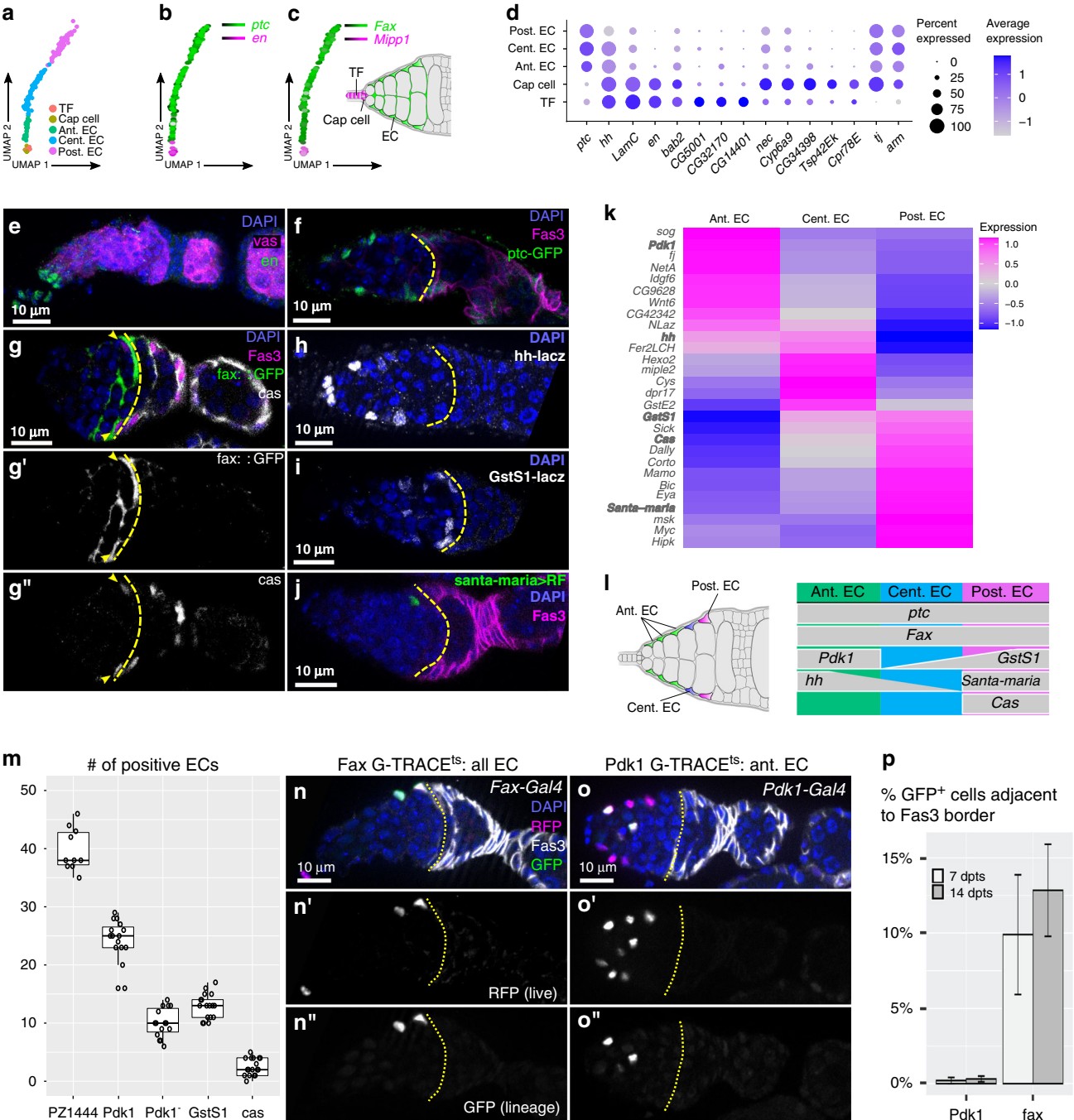

**Fig. 3 Anterior germarial somatic cells. a–c** UMAP plots of the five clusters that contain somatic cells in the anterior half of the germarium (**a**) and markers that distinguish apical cells (cap cells and terminal filament cells) from escort cells (**b–c**). **d** Dot plot showing the expression of selected markers in each of the five clusters. **e–j** Wildtype germarium stained for en (**e**), or germaria from enhancer traps or protein traps stained for GFP, RFP, or LacZ as indicated (**f–j**). fax::GFP germaria were also stained for cas (white, **g″**), the GFP channel is shown separately (white, **g′**). Staining for en, GFP or RFP (green), LacZ (white), Fas3 or vasa (magenta) and DAPI (blue). Yellow dotted line demarks the Region 2a/2b border. **k–l** Heat map (**k**) and summary diagram (**l**) showing markers distinguishing aECs, cECs, and pECs. **m** Quantification of the number of ECs per germarium that express the indicated marker genes. Each dot is a germarium. $n = 10, 15, 15, 17, 17$ germaria for PZ1444, Pdk1, Pdk1⁻, GstS1, and cas, respectively. In the box plots, the midline corresponds to the median; the lower and upper hinges correspond to the first and third quartiles; and the whiskers span the smallest and largest values within 1.5 of the interquartile range. **n–o** Germaria from flies with fax-Gal4 (**n**) or Pdk1-Gal4 (**o**) combined with G-TRACE$^{ts}$ raised at 18 °C, shifted to 29 °C upon eclosion, and well-fed for 14 days before dissection, stained for GFP (green), RFP (magenta), Fas3 (white), and DAPI (blue). **n′, o′** RFP channel (white). **n″, o″** GFP channel (white). Yellow dotted line demarks the Region 2a/2b border. **p** Quantification of the percent of germaria with GFP⁺ ECs adjacent to the Region 2a/2b border (Fas3 expression boundary) in germaria with fax-Gal4 or Pdk1-Gal4 driving G-TRACE$^{ts}$ after 7 or 14 days at 29 °C. $n = 380, 865, 81,$ and 187 GFP⁺ cells adjacent to Fas3 border for Pdk1 7dpts, Pdk1 14 dpts, fax 7dpts, and fax 14dpts, respectively. Error bars indicate S.E.M. TF: terminal filament; EC: escort cell; ant.: anterior; cent.: central; post.: posterior.

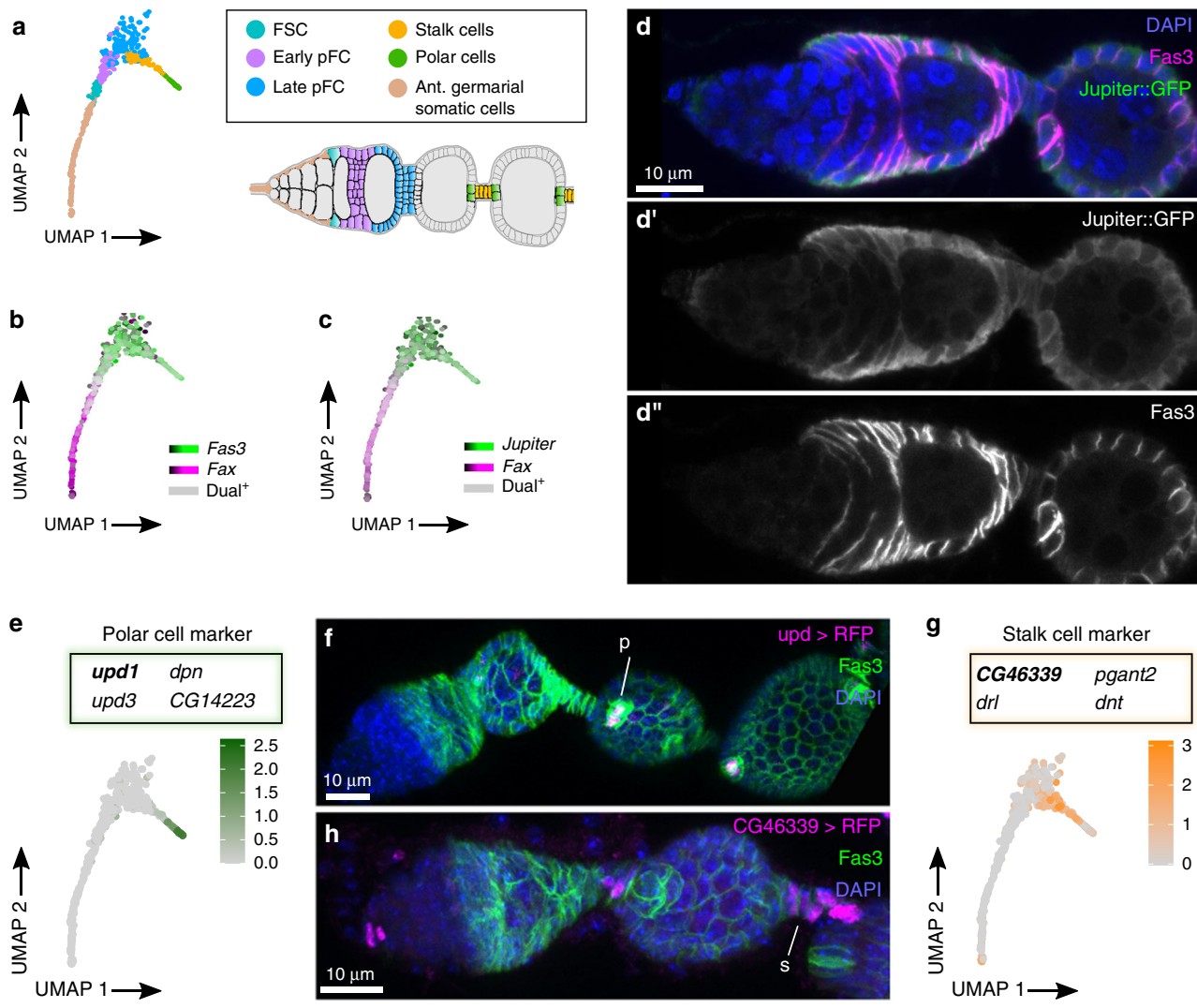

**Fig. 4 Posterior germarial somatic cells, polar cells, and stalk cells. a–c** UMAP plots of the clusters that contain the cells in the early FSC lineage, polar cells, and stalk cells showing the distribution of clusters (**a**), and the expression patterns of *Fas3* (**b**) and *Jupiter* (**c**), which are strongly expressed in these clusters, relative to *fax*, which is a marker of ECs. **d** Germarium with Jupiter::GFP stained for GFP (green), Fas3 (magenta), and DAPI (blue). **d′** GFP channel shown in white. **d″** Fas3 channel shown in white. **e–h** UMAP plots showing the specificity of *upd1* expression in the polar cell cluster (**e**) and *CG46339* expression in the stalk cell cluster (**g**) and germaria with *upd1-Gal4* (**f**) or *CG46339-Gal4* (**h**) driving RFP expression stained for RFP (white), Fas3 (green), and DAPI (blue) shown in maximum intensity projections. The expression patterns of these enhancer trap lines are consistent with the prediction that *upd1* is expressed specifically in polar cells (p) and *CG46339* is expressed specifically in stalk cells (s). FSC: follicle stem cell; pFC: prefollicle cell; ant.: anterior.

largely confined to the RFP+ region (Fig. 3o, Supplementary Fig. 5n). In comparison, *13C06-Gal4* or *c587-Gal4* driving G-TRACE[ts] produced both EC clones and FSC clones, as expected (Supplementary Fig. 5o–q)[36,43]. To further describe the differences in the clonal patterns in *Pdk1-Gal4* and *fax-Gal4*, we looked specifically at the ECs that are adjacent to the boundary of Fas3 expression. We found that only 0.3% (*n* = 865) of the GFP+ cells in *Pdk1-Gal4* germaria were adjacent to the Fas3 border, whereas 13.4% (*n* = 187) of GFP+ cells in *fax-Gal4* germaria were in this position (Fig. 3n–p). Taken together, these data indicate the aECs do not intermingle with the cEC and pEC populations, and all ECs, including those that are adjacent to the Fas3 border, do not typically contribute to the FSC lineage.

**The early follicle cell lineage**. Four terminal clusters on the same branch as ECs, TF cells, and cap cells express follicle cell markers such as *Fas3* and *Jupiter* (Fig. 4a–d)[44,45], and thus are part of the FSC lineage. One contains polar cells, as indicated by the strong expression of *unpaired1* (*upd1*)[46] (Fig. 4e, f) and another contains

stalk cells, as indicated by expression of the novel marker, *CG46339* (Fig. 4g). We found that *CG46339-Gal4* is expressed specifically in stalk cells, but RNAi knockdown of *CG46339*, which encodes for an aminopeptidase, is not sufficient to impair the formation of interfollicular stalks (Fig. 4h, Supplementary Fig. 6a).

The remaining two *Fas3*+, *Jupiter*+ clusters in this Tier 1 cluster do not express markers of the mature polar and stalk cell state and thus contain the FSCs and pFCs. To obtain increased resolution into the transcriptional differences between these clusters, we combined them and performed monocle3 (Supplementary Fig. 6b–d, Supplementary Data 10). This analysis sorted the cells in the two clusters to opposite ends of the pseudotime trajectory (Fig. 5a, pink and blue lines and Supplementary Fig. 6d), suggesting that one cluster contains cells in an earlier stage of differentiation than the other. The cells at the beginning of pseudotime express genes such as *chickadee* (*chic*) that are known to be expressed in cells at the Region 2a/2b border[44,47] and several novel markers, including *GstS1* and *CG9674*, which encodes a glutamate synthase (Fig. 5a, Supplementary Data 9).

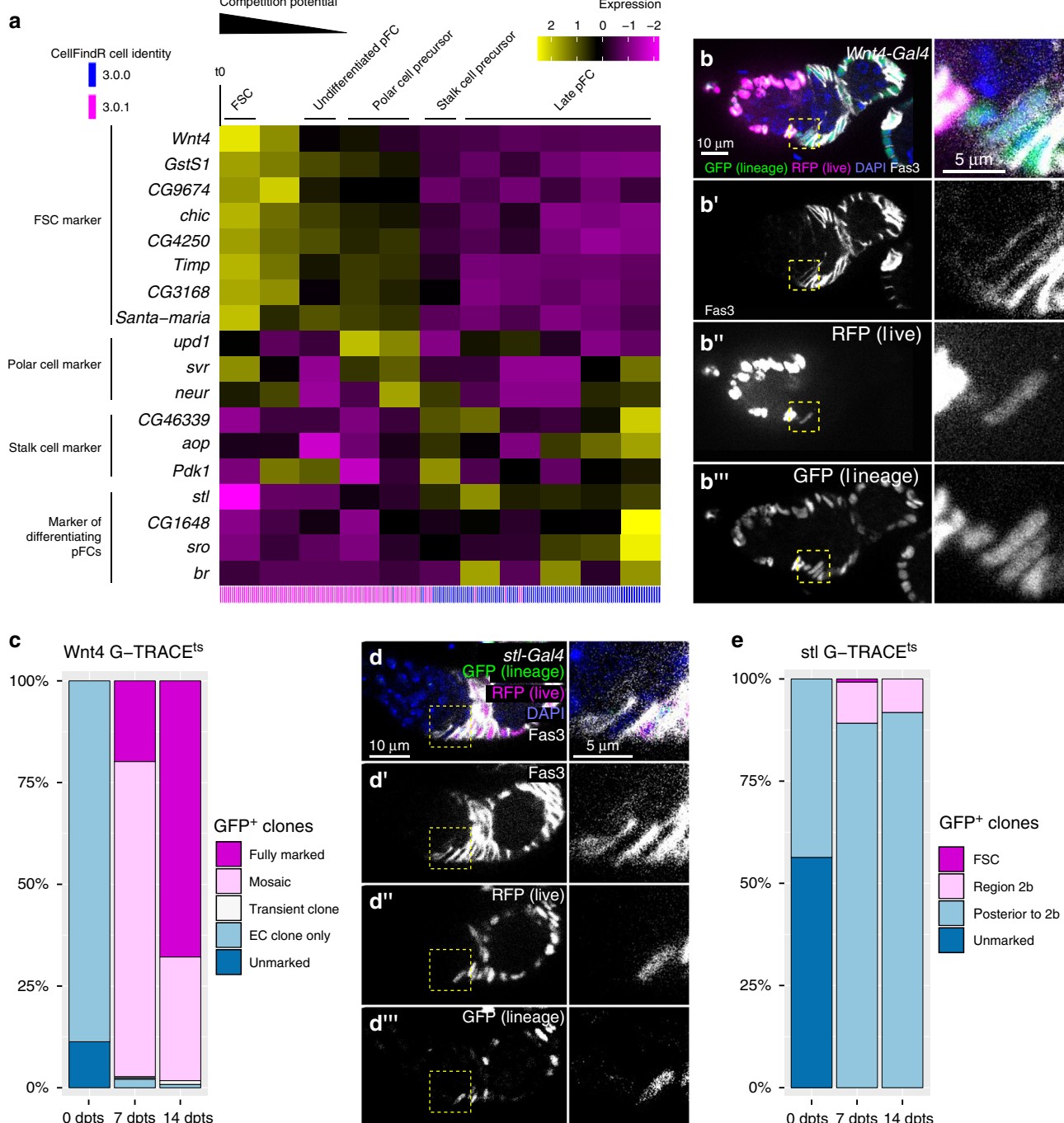

**Fig. 5 The early FSC lineage. a** Heat map showing gene expression across pseudotime in the FSC and pFCs clusters. Blue and magenta lines indicate the original CellFindR identity. **b** Germarium with *Wnt4-Gal4* driving G-TRACE[ts] stained for RFP (magenta), GFP (green), Fas3 (white), and DAPI (blue). Inset shows *Wnt4-Gal4*[low] Fas3[+] cell at the Fas3 boundary. Fas3 (**b'**), RFP (**b''**) and GFP (**b'''**) are also shown in white. **c** Quantification of ovarioles with *Wnt4-Gal4* driving G-TRACE[ts] without GFP-positive cells (unmarked), with only EC clones, transient follicle cell clones, mosaic labeling of the follicle epithelium or a fully marked follicle epithelium at 0, 7, or 14 days post temperature shift (dpts). The presence of ovarioles with EC clones at the 0 dpts time point is likely because this is where *Wnt4-Gal4* activity is strongest and Gal4 may not be fully repressed by Gal80 in these cells. Notably, we never observed GFP[+] follicle cell clones at 0 dpts, consistent with lower expression of *Wnt4-Gal4* in FSCs. *n* = 149, 156 and 120 ovarioles for 0, 7 or 14 dpts respectively. **d** Germarium with *stl-Gal4* driving G-TRACE[ts] stained for RFP (magenta), GFP (green), Fas3 (white), and DAPI (blue). *stl-Gal4* drives RFP expression sparsely in pFCs in the 2b Region and is consistently expressed in Region 3. GFP[+] clones typically include pFCs in region 2b (inset) but not FSCs or pFCs at the 2a/2b border. Fas3 (**d'**), RFP (**d''**) and GFP (**d'''**) are shown separately in white. **e** Quantification of ovarioles with *stl-Gal4* driving G-TRACE[ts] without GFP-positive cells (unmarked), with FSC clones, transient follicle cell clones located in Region 2b, or transient follicle cell clones posterior to Region 2b at 0, 7 or 14 dpts. Ovarioles at the 0 dpts frequently contained small GFP[+] clones of up to 4 cells posterior to region 3. These clones were usually confined to stalk cells where *stl-Gal4* activity is strongest. *n* = 140, 135 and 128 ovarioles for 0, 7 or 14 dpts respectively. FSC: follicle stem cell; pFC: prefollicle cell; EC: escort cell; dpts: days post temperature shift.

We found that enhancer traps of both genes are expressed in Fas3[+] cells near the Region 2a/2b border but not in pFCs located further to the posterior (Supplementary Fig. 6e–f). This confirms that these early stages of pseudotime identified by monocle3 correspond to the earliest stages of the FSC lineage.

Our analysis predicted that *Wnt4* is expressed in ECs and the early FSC lineage (Fig. 5a, Supplementary Fig. 6g) and indeed, we observed that *Wnt4-Gal4* driving G-TRACE[ts] produced strong RFP expression in all ECs and significantly lower RFP expression in just $2.08 \pm 0.8$ cells per germarium ($n = 79$ germaria). Nearly all of the *Wnt4-Gal4*[low] cells (97.7%, $n = 130$ cells) were at the edge of the *Fas3* expression boundary where the FSCs are expected to reside (72.3% of *Wnt4-Gal4*[low] cells in this position were Fas3[+]; 25.4% were Fas3[−] Fig. 5b)[37,44,48]. In the GFP channel, we observed large FSC clones that extended through the germarium and across multiple follicles (and thus must have originated from an FSC) in over 97% of the ovarioles at 7 and 14 dpts, including many in which all of the follicle cells in the ovariole were GFP[+] (Fig. 5b, c). Lastly, we assayed for GFP expression in Wnt4::GFP germaria and detected GFP[+] puncta in Fas3[+] cells at the boundary of Fas3 expression, confirming that Wnt4 protein is expressed in these cells (Supplementary Fig. 6h). It is unclear whether all Fas3[+] cells at the border of Fas3 expression are FSCs, but these results indicate that Wnt4 is expressed in FSCs and that FSCs are typically *Wnt4-Gal4*[low] whereas ECs to the anterior of the Fas3 expression boundary are *Wnt4-Gal4*[high], as expected[49–51] and pFCs to the posterior of this boundary are typically *Wnt4-Gal4*[off].

The next stage of pseudotime contains cells that do not express high levels of FSC markers but have not yet begun to upregulate markers of differentiation, suggesting that they are the early pFCs just downstream from the FSC state. The first type of differentiation to appear in pseudotime are the polar cell precursors, which are characterized by the upregulation of markers such as *upd1* and *neuralized* (*neur*) (Fig. 5a). Consistent with this, polar cell differentiation is the earliest cell fate decision made by pFCs[32,48,52,53]. Polar cells induce neighboring pFCs to differentiate into stalk cells and, indeed, the next stage of differentiation in pseudotime contains stalk cell precursors, which express the stalk cell markers *CG46339*, *Pdk1*, and *anterior open* (*aop*)[54] (Fig. 5a, Supplementary Fig. 6i–j). Polar cell differentiation begins in Region 2b[48], and the onset of polar cell differentiation in pseudotime marks the transition from the early pFC cluster (pink lines) to the late pFC cluster (blue lines). This suggests that the early pFCs reside mainly in Region 2b while late pFCs begin in Region 3.

The onset of the late pFC stages in pseudotime is marked by a peak in the expression of *stall* (*stl*) and the expression pattern of *stl-Gal4* is consistent with this prediction. Specifically, we found that *stl-Gal4* driving G-TRACE[ts] produced RFP expression sporadically in Region 2b pFCs and consistently in Region 3 pFCs, but never in Fas3[+] cells at the Region 2a/2b border, where the FSCs reside (Fig. 5a, d, e). Interestingly, although *stl-Gal4* driving G-TRACE[ts] occasionally produced GFP positive pFCs clones in Region 2b (4.9% germaria at 7d and 7.8% germaria at 14d contain GFP[+] pFCs), it rarely produced GFP[+] FSCs (0.4% germaria at 7d and 0% at 14d contain FSC clones), suggesting that pFCs located even just one or two cell diameters downstream from the Fas3 border do not normally participate in FSC replacement events. The next stages of pseudotime are characterized by a wave of transcriptional changes, and one of the latest markers to peak in expression is *broad* (*br*). We find that a GFP trap in the Z2 domain (*br*[Z2]-GFP) exhibits expression in Region 3/Stage 1 pFCs but is not detectable in main body follicle cells in Stage 2, suggesting that the pseudotime trajectory ends with Region 3/Stage 1 follicle cells (Fig. 5a, Supplementary Fig. 6k).

Interestingly, many genes that monocle3 predicts are upregulated in FSCs relative to pFCs are also highly expressed in one or more EC populations, and the overall transcriptional profile of pECs is particularly close to the transcriptional profiles of FSCs and pFCs (Fig. 6a–c). These findings are consistent with a common developmental origin of ECs and FSCs[55], and suggest that a single marker that distinguishes FSCs from both ECs and pFCs may be rare. This may explain why many genes that are predicted to distinguish FSCs from pFCs are not well suited to discriminate between FSCs and ECs (Fig. 6a).

The early and late pFC clusters are distinguished from each other by opposing gradients of *zfh1* and *stl* expression (Fig. 6c, Supplementary Fig. 6l) and, indeed, we observed corresponding gradients in vivo. Specifically, we found that *zfh1* is expressed strongly in Region 2b pFCs and tapers off in Region 3 pFCs whereas *stl* expression becomes more uniform starting in Region 3 pFCs, as described above (Figs. 5d, 6d, Supplementary Fig. 6m). In addition, the cells in the early pFC cluster generally co-express *cas* and *eya* whereas cells in the late pFC cluster start to segregate into *cas*[high], *eya*[low] or *cas*[low], *eya*[high] states (Fig. 6e, f), which is an indication of differentiation[52]. Together, these observations provide a set of markers that distinguish the FSCs, early pFCs, and late pFCs (Fig. 6f, g). In addition, our lineage tracing experiment in combination with other studies[56] demonstrates that while FSCs can be replaced by pFCs, not all pFCs are fit for competition. This provides evidence for heterogeneity among these transit-amplifying cells of the follicle cell lineage.

**MB follicle cell transcriptomes change in time and position.** The remaining terminal clusters contain main body follicle cells and their derivative cell types (Fig. 7a–d). These clusters lack *zfh-1*[+] cells (Supplementary Fig. 7a), indicating that they correspond to Stage 2 and later. Follicle cell differentiation in these stages is a continuum, with relatively homogeneous populations of main body follicle cells in early stage follicles and more diverse populations in mid and late-stage follicles[57]. Accordingly, main body follicle cell clusters are distinguished by stage-specific markers as well as positional markers expressed in subsets of mid and late-stage follicle cells (Fig. 7e). One cluster strongly expresses *Fasciclin 2* (*Fas2*) and *N-cadherin* (*CadN*), which are expressed in main body follicle cells[58,59] from the germarium to approximately Stage 6, but has very few cells expressing markers such as *broad* (*br*) which is first detected in Stage 5[60] (Fig. 7f–j, Supplementary Fig. 7b–d). This indicates that it primarily contains Stage 2–5 main body follicle cells. Other terminal clusters express *br*, but not *Yp1*, which is a marker of vitellogenesis[61] and first becomes detectable in Stage 7 follicles (Fig. 7h–k, Supplementary Fig. 7d–e), thus placing them in the Stage 5–6 range. Starting in Stage 5, main body follicle cells begin to exhibit regional specialization along the anterior/posterior axis. A subset of cells in one of the *br*[+] *Yp1*[−] cluster expresses the marker *mirror* (*mirr*), which is marks central follicle cells starting in Stage 6[62] while other cells in the same cluster do not express any stage-specific markers yet, suggesting that this cluster contains both anterior and central follicle cells of the Stages 5–6 (Fig. 7l, m, Supplementary Fig. 7f). The second *br*[+], *Yp1*[−] cluster expresses the posterior follicle cell markers, *midline* (*mid*) or *pointed* (*pnt*)[54,63] (Fig. 7l–n, Supplementary Fig. 7g–h). Posterior and central cells from Stage 7 or later express *Yp1* and can be distinguished from each other by the expression of *pnt* and *mid* or *mirr* (Fig. 7i, l, m, Supplementary Fig. 7e–h).

At these stages, the anterior follicle cells begin to acquire a stretch cell identity. We found that the stretch cells and their Stage 6 precursors segregated into two terminal clusters that express the stretch cell marker, *dpp*[64], and *cv-2*, which encodes a dpp binding

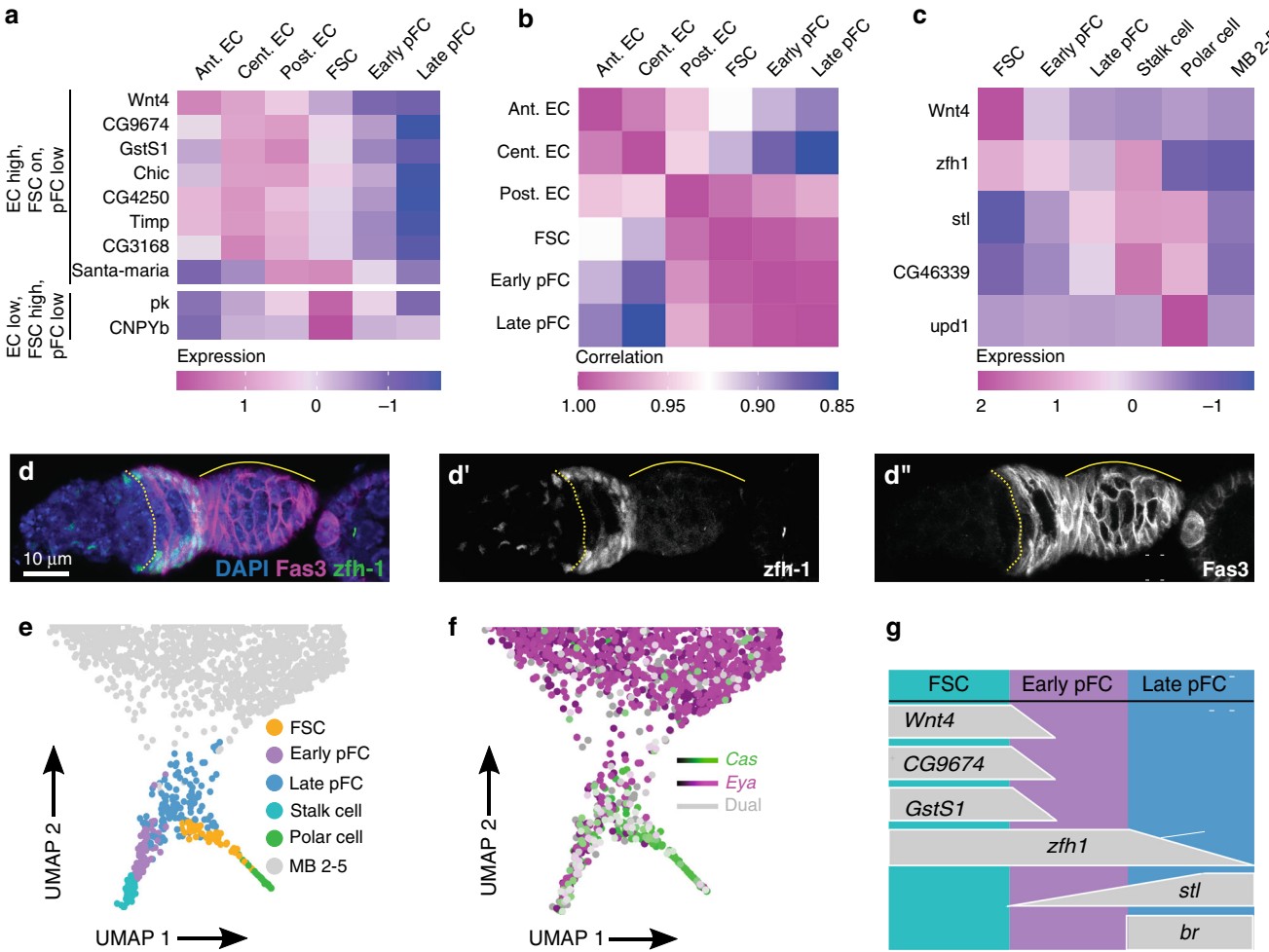

**Fig. 6 Distinct expression patterns in cell types of the early follicle cell lineage. a** Heat map of gene expression in EC populations, FSCs and pFCs. Most stem cell markers identified by pseudotime analysis of the early FSC lineage (Fig. 5a) are also expressed in ECs. In contrast, *pk* and *CNPYb* show high expression on FSCs but not ECs and pFCs. **b** Heat map showing the correlation in the gene expression profiles of EC, FSC and pFC clusters. We observe a high degree of similarity between cell types of the same lineage, as expected, but also find that the gene expression profile of pECs has substantial overlap with FSCs and pFCs. **c** Heat map showing the expression of markers that can be used to distinguish FSCs, early pFCs, late pFCs and stalk and polar cells from each other. **d** Maximum intensity projection of a wildtype germarium stained for zfh1 (green), Fas3 (magenta), and DAPI (blue). **d'** zfh1 staining (white). **d''** Fas3 staining (white). **e–f** UMAP plots showing FSC, pFC, stalk and polar cell clusters (**e**) and the expression patterns of *cas* and *eya* (**f**). **g** Summary of markers that can be used to distinguish FSCs, early pFCs, and late pFCs from each other. FSC: follicle stem cell; pFC: prefollicle cell; EC: escort cell; ant.: anterior; cent.: central; post.: posterior; MB: main body follicle cell.

partner[65] and is a novel marker of this cell population (Fig. 7l, m, o–q, Supplementary Fig. 7i–j). *br* is expressed in stretch cells until Stage 8 (Fig. 7r, Supplementary Fig. 7d), and one stretch cell cluster contains *br+* cells throughout (Fig. 7e), while the other contains only a small subset of *br+* cells but is mostly *br−*. Instead, this cluster highly expresses *Vha16-1* (Fig. 7e), which is expressed in stretch cells starting at Stage 10 to induce nurse cell death[66]. In addition, this cluster also contains a subset of cells that express *Sox14*, which we found is expressed in late-stage follicles beginning in Stage 9 (Fig. 7i, t, Supplementary Fig. 7k). This suggests that one cluster contains the stretch cell precursors and early stretch cells (~Stage 6-8) while the other contains the more mature (~Stage 8+) stretch cells. A subset of cells in the Stage 6-7 stretch cell cluster expressed *slow border cells* (*slbo*), which is highly expressed in border cells[67], so we manually segregated these cells into a separate border cell cluster (Fig. 7b, e). We found that the cells in this cluster are distinguished by several unique markers (Supplementary Fig. 8a–d). The *slbo-Gal4* enhancer trap line is expressed in border cells and posterior follicle cells[68], and we found that the top 100 most upregulated genes in these cell types[69]

aligned well with the corresponding clusters in our dataset (Supplementary Fig. 8e). The remaining three clusters contain the late-stage somatic cells and are distinguished by high levels of *Yp1* expression and *mirr* in the Stage 8 central follicle cells; high *Yp1*, *pnt*, and *mid* in Stage 8 posterior follicle cells; and *Yp1* and *Sox14* in the Stage 9+ follicle cells (Fig. 7e, i, l, m, Supplementary Fig. 7e–h, k).

To assay for transcriptional changes that occur during follicle development, we applied monocle3 (Supplementary Fig. 8f–h). This analysis placed the cells from early-stage follicles that express high levels of *CadN* and *Fas2*, at one end, and cells from late-stage follicles that express high levels of *Sox14* at the other end (Fig. 7s). In addition, monocle3 made accurate predictions about the stage-specific expression of several other genes. For example, it predicted that *Fas3* and *SPARC* expression decrease in early stages of follicle development (Fig. 7s) and, indeed, we found that expression of Fas3 and SPARC both tapered off by Stage 3–5 (Fig. 7u, v, Supplementary Fig. 7l–m), consistent with previous findings[46,70]. We searched for stage-specific transcription factors by comparing genes in the GO-term for transcription regulator

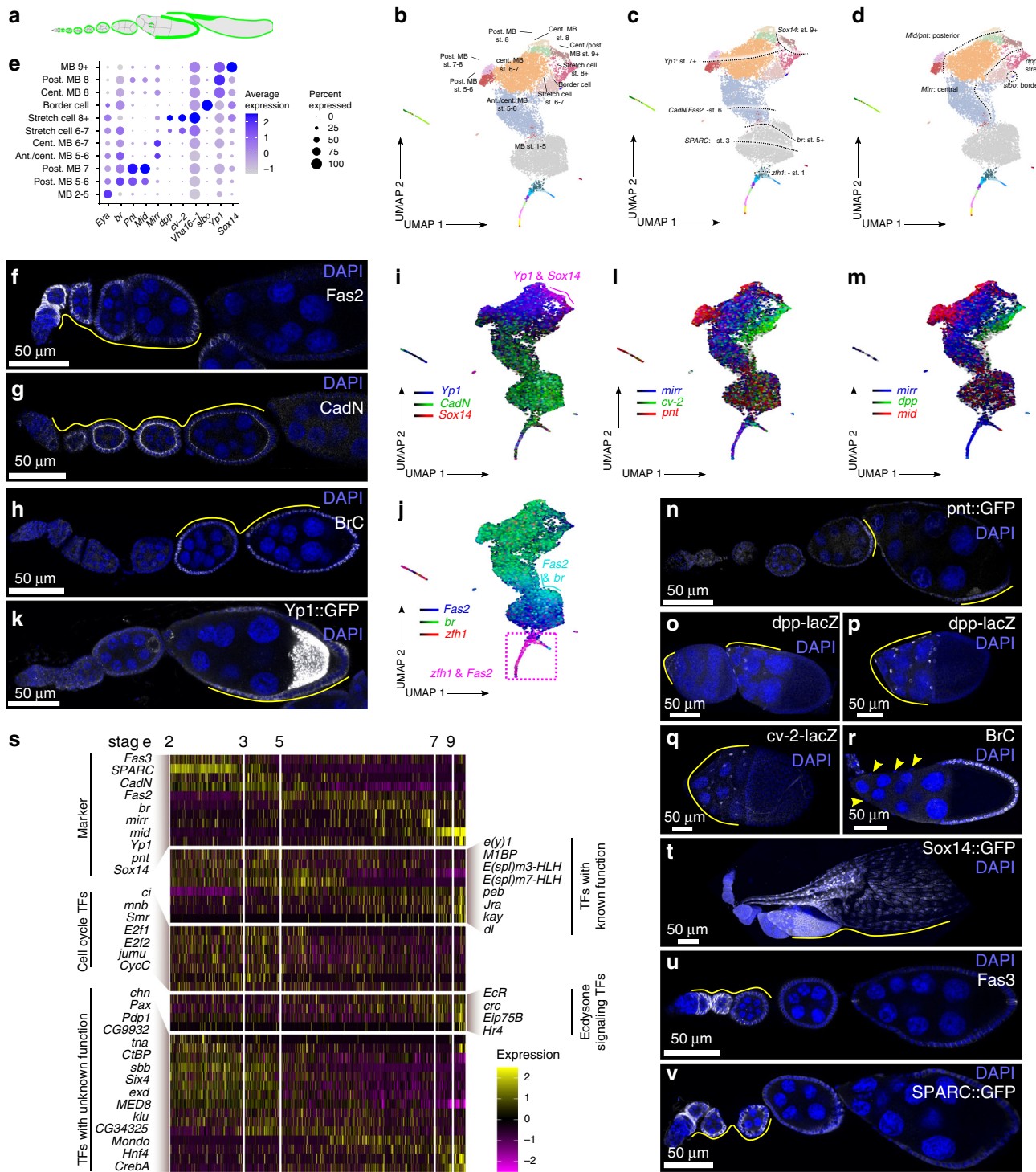

**Fig. 7 Distinct stages of main body follicle cells. a–d** A diagram of the ovariole highlighting the main body follicle cells in green (**a**), and UMAP plots showing main body follicle cell cluster identities (**b**). The spatial arrangement of cells on the plot positions cells according to the stage of oogenesis (**c**) and, starting in mid-oogenesis, anterior-posterior position on the follicle (**c**). **e** A dot plot showing the expression patterns of genes that are expressed in different types of main body follicle cells and their derivatives. **f–r** The expression patterns of selected markers on UMAP plots or in ovarioles, as indicated. In all images of ovarioles, the selected marker is shown in white and DAPI is shown in blue. The images in (**f**, **g**, **h** and **r**) show wildtype ovarioles stained for Fas2 (**f**), CadN (**g**), and BrC (**h** and **r**). The remaining images are of germaria from enhancer trap or protein trap lines, as indicated. Maximum intensity projections are shown in (**o–q**). **s** Heat map showing the changes in gene expression across pseudotime in main body follicle cells. Genes that serve as markers of different stages of pseudotime as well as transcription factors with predicted functions, such as cell cycle genes at the early stages before the mitosis-to-endocycle switch, and ecdysone responsive genes at late stages are shown. Transcription factors with unknown functions in oogenesis that have dynamic expression patterns across pseudotime are also shown. **t–v** Ovarioles from Sox14::GFP (**t**); wildtype (**u**) or SPARC::GFP (**v**) flies stained for GFP or Fas3 (white) and DAPI (blue). MB: main body follicle cell; ant.: anterior; cent.: central; post.: posterior; TFs: transcription factors.

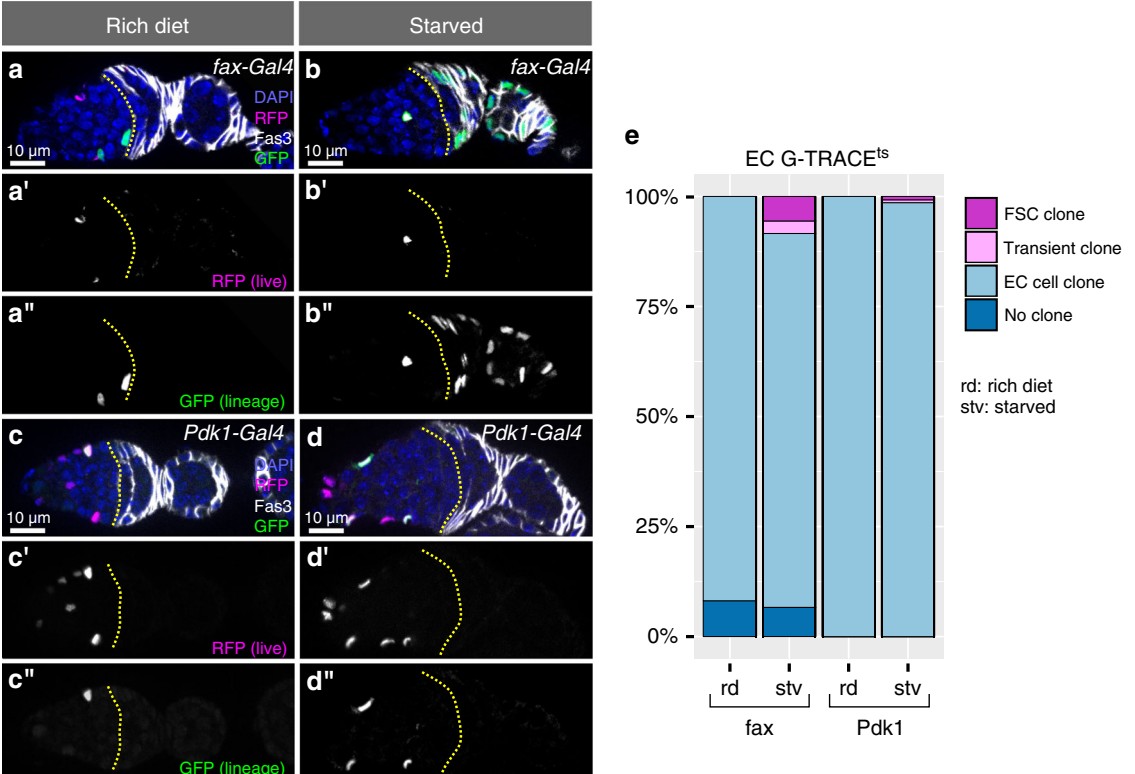

**Fig. 8 Posterior ECs convert to FSCs in response to nutrient deprivation. a–d** Germaria from flies with fax-Gal4 (**a**, **b**) or Pdk1-Gal4 (**c**, **d**) driving G-TRACE[ts] stained for DAPI (blue), Fas3 (white), RFP (magenta), and GFP (green). **a′–d′** RFP channel in white. **a″–d″** GFP channel in white. Yellow line outlines the 2a/2b border. fax-Gal4 and Pdk1-Gal4 do not produce GFP + FSC clones on a rich diet (0 out of 5 flies with follicle cell clones) (**a**, **c**), whereas, following exposure to 24 h of starvation, fax-Gal4-expressing cells produce GFP + FSC clones (4 out of 5 flies contained follicle cell clones) (**b**) but Pdk1-Gal4-expressing cells almost never do (**d**) (2 out of 5 flies with follicle cell clones, we observed 1 ovariole with a transient clone and 1 ovariole with an FSC clone out of 138 ovarioles total). **e** Quantification of GFP + clone types in flies with *fax-Gal4* or *Pdk1-Gal4* driving G-TRACE[ts] exposed to a rich diet or starvation. $n = 170$, 211, 134, and 138 ovarioles for *fax-Gal4* rich diet, *fax-Gal4* starved, *Pdk1-Gal4* rich diet, and *Pdk1-Gal4* starved, respectively. p-values from two-sided Student's *T*-test for comparisons of the frequency of follicle cell clones (FSC + transient): *fax-Gal4* rich vs starved: $p = 0.002$; *Pdk1-Gal4* rich vs starved: $p = 0.57$. EC: escort cell; rd: rich diet; stv: starved; FSC: follicle stem cell.

activity with genes identified by monocle3 to be differentially regulated in follicle cell pseudotime (Supplementary Data 11) and identified 363 genes that fit these criteria. Among these we identified several transcription factors with known roles in oogenesis, such as cell cycle regulators in early stage follicles and ecdysone responsive transcription factors in late stages as well as others with unknown functions in oogenesis (Fig. 7s).

**A subpopulation of ECs can convert to FSCs under starvation.** The identification of Gal4 lines that are expressed in subsets of ECs provided us with the opportunity to investigate functional differences among cells in the EC population. As described above, *fax-Gal4* is expressed sporadically throughout the EC population and generally did not produce G-TRACE[ts] clones in the follicle epithelium (Fig. 3n, Supplementary Fig. 5m). However, our observation that follicle cell clones were present in the ovarioles from one fly prompted us to consider whether environmental conditions such as nutrient availability could affect the pattern of clone formation. We found that, with the EC driver *fax-Gal4* driving G-TRACE[ts], exposure to 24 h of total starvation (water only) or protein starvation (water plus sucrose) produced FSC clones in 80% ($n = 5$) or 71% ($n = 7$) of flies examined, respectively. In total, we found that an average of 8.4% ($n = 211$) of ovarioles from flies exposed to total starvation and 6.1% ($n = 185$) of ovarioles from flies exposed to protein starvation contained FSC or transient follicle cell clones (Fig. 8a, b, e, Supplementary

Fig. 9a). In contrast, we did not find any follicle cell clones when flies were kept on a rich diet for the same period of time ($n = 5$ flies, $n = 170$ ovarioles) (Fig. 8a, b, e). Total starvation did not expand the expression of RFP into the Fas3[+] region (Supplementary Fig. 9b–d), indicating that the emergence of clones is not due to the expression of *fax-Gal4* in follicle cells. Consistent with this, we confirmed previous reports[35] that these starvation conditions cause a decrease in fax::GFP expression in ECs (Supplementary Fig. 9e–g). As an additional test, we shifted flies to 18 °C to inactivate the clone induction capability of G-TRACE[ts] before exposing flies to total starvation. Indeed, we found that FSC clones emerged in the starved group but not in paired controls on a rich diet in these conditions as well, albeit at a lower frequency (Supplementary Fig. 9a).

With the aEC driver *Pdk1-Gal4*, we found that flies exposed to identical starvation conditions produced FSC or follicle cell G-TRACE[ts] clones in only 1.4% of ovarioles ($n = 138$) (Fig. 8d, e). *Pdk1-Gal4* produces many more GFP[+] ECs per germarium than *fax-Gal4* ($14.8 \pm 4.6$, $n = 60$ for *Pdk1-Gal4*; $2.5 \pm 1.5$, $n = 73$ for *fax-Gal4*) but only rarely produces GFP[+] EC clones in Region 2a (Fig. 3n–p, Supplementary Fig. 5n). Together, this suggests that the ECs located in Region 2a can convert to FSCs under starvation conditions while those located in Region 1 cannot. We reasoned that only germaria with a GFP[+] EC on the Region 2a/2b border could display FSC clones after starvation. Therefore, we considered these germaria separately. In well-fed conditions, germaria with *fax-Gal4* driving G-TRACE[ts] did not contain FSC

clones irrespective of the GFP+ EC position ($n = 45$ germaria with ECs in Region 1, 20 germaria with ECs in Region 2a). In contrast, 29.7% ($n = 37$) of germaria with GFP+ ECs in Region 2a contained clones upon starvation, while only 6% of germaria with labeled ECs in Region 1 contained FSC clones ($n = 134$). Taken together, these observations strongly suggest that the ECs along the Region 2a/2b border, which are primarily pECs, but not more anteriorly located aECs, are able to convert to FSCs in response to starvation. To identify genes that regulate the conversion of ECs to FSCs, we performed a candidate screen for genetic perturbations that induce FSC clones with *fax-Gal4* driving G-TRACE[ts] in well-fed conditions. We observed the emergence of FSC clones with RNAi-knockdown of *escargot* (*esg*), which also causes niche cell conversion in the *Drosophila* testis[71,72], overexpression of *Rheb*, which is an activator of mTOR signaling, or overexpression of a constitutively active allele of *Toll* (*Tl*) (Fig. 9a–d). Since these genetic perturbations were limited to *fax-Gal4* expressing cells, these observations provide additional confirmation that the FSC clones originate from EC conversion events and also demonstrate that the process is genetically controlled (Fig. 9e).

## Discussion

In summary, we have generated a detailed atlas of the cells in the adult *Drosophila* ovary. This atlas consists of 26 clusters that each correspond to a distinct population in the ovary. Through experimental validation and referencing well-characterized markers in the literature, we determined the identity of each cluster, and found that all of the major cell types in the ovariole are represented. We further identified several transcriptionally distinct subpopulations within these major cell types, such as the anterior, central, and posterior EC populations. We also identified both the GSCs and the FSCs in our dataset, which revealed several genes that are predicted to be specific for each of these stem cell populations. In addition, we identified several Gal4 drivers, including *Pdk1-Gal4*, *fax-Gal4*, and *stl-Gal4*, with unique expression patterns that make it possible to target transgene expression to the subsets of cells marked by these drivers. Lastly, although we have primarily focused on the most uniquely expressed genes for each cluster in this study, the transcriptional profile of each cluster is a rich dataset that can be mined to identify populations of cells that are relevant for a topic of interest (Supplementary Data 1–5). For example, we compared the gene expression profile of each cluster to a list of human disease genes that are well-suited for analysis in Drosophila[73]. We found that germ cells are enriched for cells expressing major drivers of cancer, and ECs and follicle cells are enriched for genes involved in cardiac dysfunction (Supplementary Fig. 10), suggesting that these cell types may be good starting points for studies into the genetic interactions that underlie these human diseases.

This study also demonstrates the utility of using CellFindR[14] in combination with monocle3[25] to identify unique populations of cells within a dataset. Because CellFindR produces clusters in a structured, iterative fashion, we were able to construct a hierarchical tree that corresponds to a transcriptome relationship between clusters, and this outperformed other clustering methods. The tree built by CellFindR aligns well with expectations and provides some interesting new insights. For example, we expected that germ cells would cluster apart from somatic cells in Tier 1 because these populations are substantially different from each other, arising at different times during development and from completely different lineages. However, it was surprising that the FSC, pFCs, polar cells, and stalk cells clustered more closely to ECs than to the follicle cells of budded follicles. This suggests that many cell types in the germarium, which are often studied separately, have biologically relevant similarities.

Our use of G-TRACE to assess the lineage potential of somatic cells in the germarium generated several insights. First, it provided support for the view that FSCs reside within the Fas3 expression boundary and express low levels of Fas3[3,4], rather than anterior to the Fas3 expression boundary, as proposed recently[5–7]. Specifically, we found that *Wnt4-Gal4* driving G-TRACE produced FSC clones with very high frequency whereas *fax-Gal4* driving the same G-TRACE construct typically did not, at least under well-fed conditions. Analysis of RFP expression in these ovarioles indicated that *Wnt4-Gal4* is expressed at low levels in the weakly Fas3+ cells at the Fas3 expression boundary whereas *fax-Gal4* is not expressed in these cells, suggesting that expression within the Fas3+ population is required to produce FSC clones in well-fed conditions. Second, further analysis of the flies with *fax-Gal4* driving G-TRACE led to the surprising finding that ECs can convert to FSCs under starvation conditions. Recent studies have described similar forms of cellular plasticity in other tissues[71,72,74–77], suggesting that the ability of non-stem cells to convert to stem cells may be a more general feature of adult stem cell niches. However, this aspect of tissue homeostasis remains poorly understood. Our finding that the conversion of ECs to FSCs can be induced by perturbations of mTor or Toll signaling is consistent with a role for these pathways in responding to starvation and cellular stress in other tissues[78–80], and provides a new opportunity to investigate the mechanisms of cellular responses to physiological stress in an adult stem cell niche.

Overall, this study provides a resource that will be valuable for a wide range of studies that use the *Drosophila* ovary as an experimental model. Additional scRNA-seq datasets provided by other studies will further increase the accuracy and resolution of the ovary cell atlas[55,57], and it will be important to follow up on the predictions of the atlas with detailed studies that focus on specific populations of cells. Collectively, these efforts will help drive discovery forward by providing a deeper understanding of the cellular composition of the *Drosophila* ovary.

## Methods

**Single-cell sequencing of the *Drosophila* ovary.** Newly hatched flies were reared on standard lab conditions and fed wet yeast for three consecutive days. For dataset1 flies with the genotype *109-30-Gal4/+; 13CO6-GFP/UAS-CD8::GFP* were used. For datasets2 and 3 Canton-S flies were used. For datasets2 and 3 60 females were dissected within 45 min in ice cold Schneider's Insect Medium with 10% FBS and 167 mg/ml insulin on an ice pack. We enriched for the younger, non-vitellogenic stages of the ovary using micro-scissors. Tissue was transferred to an eppendorf tube containing ice-cold Cell Dissociation Buffer (Thermo Fisher Scientific #13151014) and rinsed once with the buffer. Dissociation was performed at RT in Cell Dissociation Buffer with 4 mg/ml elastase (Worthington Biochemical LS002292) and 2.5 mg/ml collagenase (Invitrogen # 17018-029) with nutation and regular pipetting with a P200 to aid tissue dissociation. After 20 min the solution was passed through a 50 µm filter (Partec #04-0042-2317) and the solution incubated for additional 10 min before passage through a 30 µm filter (Miltenyi Biotec #130-041-407). Enzymes were quenched by adding 500 µl of dissection solution and cells were centrifuged for 5 min at 4 °C and 3500 rcf. Cells were washed in dissection solution and centrifuged again before being resuspended in ice-cold 200 µl PBS with 0.04% ultrapure BSA (Thermo Fisher Scientific AM2616). Dissociation was verified and cells were counted using a cell counting chamber and the solution adjusted to 1000 cells per µl before subjection to single-cell RNA-sequencing using the Chromium Single Cell 3' Reagent Version 2 Kit (10× Genomics). Sequencing was performed on Illumina HiSeq 2500 according to the 10× Genomics V2 manual. For dataset1, ovaries from 200 flies were dissected and dissociated as described above to produce a single-cell solution. The solution of dissociated cells was subjected to MACS as described before[32]. Specifically, dissociated cells were pelleted by centrifugation at 1000 × g for 7 min at 4 °C, then resuspended in 90 µl Schneider's Insect Medium+10 µl α-CD8a MicroBeads (Miltenyi Biotec 130-049-401) per 15 flies dissected. Dissociated cells were allowed to incubate with the α-CD8 MicroBeads for 15 min at 4 °C. The CD8+ cells were then isolated by passing the cells over a magnetic column in an OctoMACS separator (Miltenyi Biotec 130- 042-108), washed with ice-cold PBS and adjusted to 1000 cells per µl with PBS with 0.04% ultrapure BSA. Cells were then subjected to single-cell sequencing using the 10X platform. For dataset1 5000 cells were loaded into one well of the 10X chip, resulting in ~500 high-quality cellular

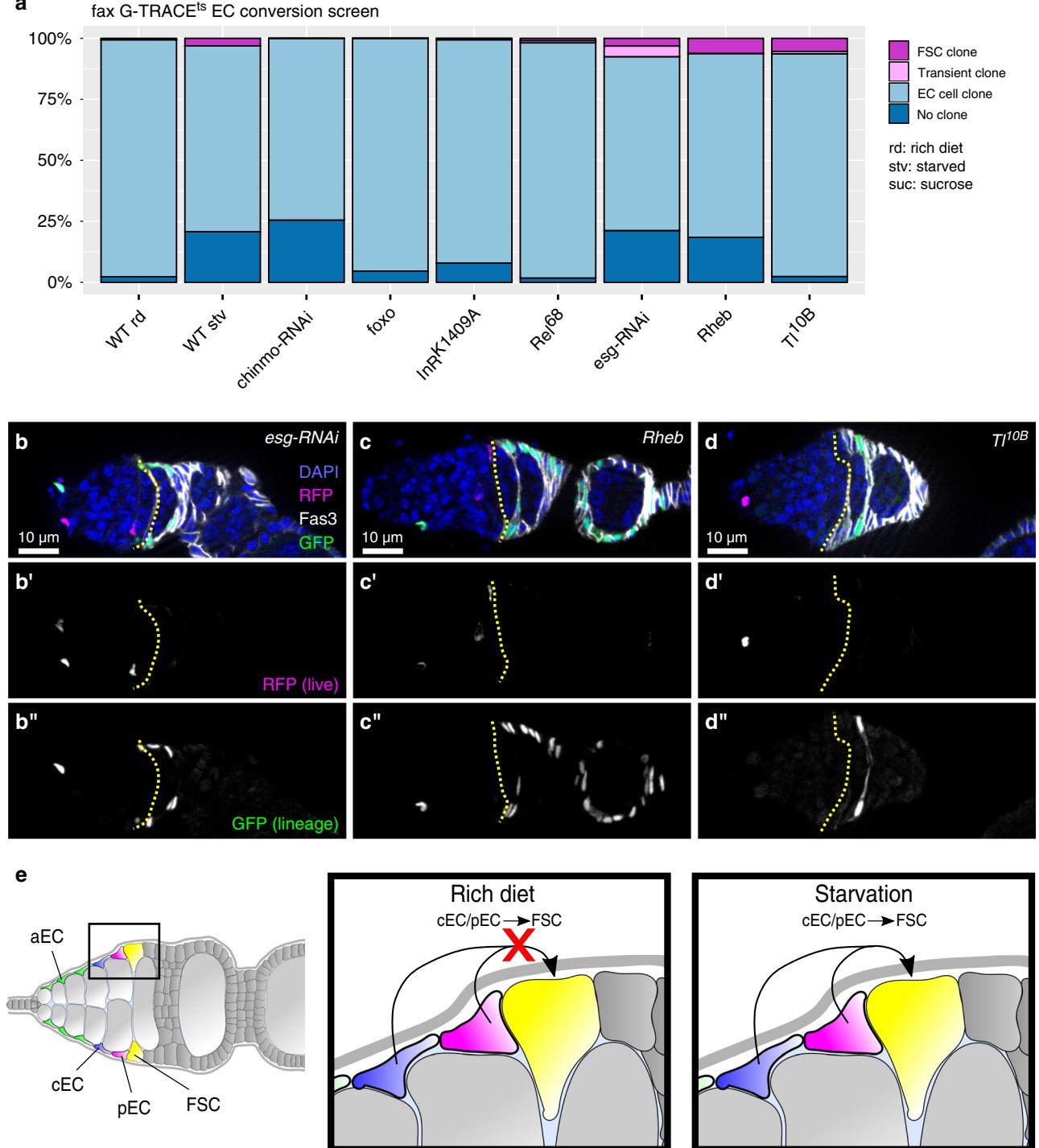

**Fig. 9 EC conversion to FSCs is genetically controlled. a** Quantification of GFP + clone types in flies with *fax-Gal4* driving G-TRACEts alone on a rich diet (rd) for 14 days (WT rich diet); starved for 24 hrs within the 14 dpts (WT starved); or on a rich diet for 14 days in combination with *chinmo* or *escargot* RNAi knockdown, or overexpression of a dominant-negative allele of the insulin receptor (*InR*[K1409A]), a constitutively active allele of *Relish* (*Rel*[68]) or *Toll* (*Tl*[10B]) or a wildtype allele of *Rheb* or *foxo*. p-values from two-sided Student's T-test for comparisons between the frequency of follicle cell clones (FSC + transient) in WT rich diet 14d and mutant condition that have a *p* < 0.05: *esg-RNAi*: *p* = 0.008; *Rheb*: p = 0.03; *Tl*[10B]: *p* = 0.03. *n* = 204 (WT rich diet), 213 (WT starved), 113 (*chinmo-RNAi*), 135 (*foxo*), 127 (*InR*[K1409A]), 125 (*Rel*[68]), 154 (*esg-RNAi*), 143 (*Rheb*), 118 (*Tl*[10B]) ovarioles. **b-d** Germaria with *fax-Gal4* driving expression of G-TRACE[ts] and *esg-RNAi* (**h**), *Rheb* (**i**), or *Tl*[10B] (**j**) stained for GFP (green), RFP (magenta), Fas3 (white) and DAPI (blue). **b'-d'** RFP channel show in white. **b''-d''** GFP channel shown in white. GFP + follicle cell clones are present even though flies were maintained on well-fed conditions. Yellow line outlines the 2a/2b border. **e** Model summarizing our observations that starvation, but not well-fed conditions, induces central ECs and/or posterior ECs to convert to FSCs. EC: escort cell; rd: rich diet; stv: starved; suc: sucrose; aEC: anterior escort cell; cEC: central escort cell; pEC: posterior escort cell; FSC: follicle stem cell.

transcriptomes. To increase the rate of captured cells, we loaded 27,000 cells for dataset2 and received ~8000 transcriptomes. In dataset3, we loaded 17,000 cells to allow capturing high numbers of cells while reducing the chance of capturing doublets and received ~5000 high-quality transcriptomes.

**Bioinformatic analysis.** Reads were aligned to the Drosophila reference genome (dmel_r6.19) using STAR v2.5.1b and resulting bam files were processed with the Cell Ranger pipeline v2.0.0 (dataset1), v2.1.1 (dataset2) or v3.1.0 (dataset3). Using Seurat v3.1.5 in Rstudio 1.2.5033 we filtered out low-quality cells based on UMI counts and the number of genes (Supplementary Table 1, Supplementary Fig. 1b-c) and removed doublets with DoubletFinder v2.0.2 and based on expression of known mutually exclusive genes using Seurat v3.1.5. We estimate ~6700 cells per ovariole, thus this dataset of ~14,000 cells achieves >2× coverage. The batch correction was performed with Seurat v3.1.5. Clustering was performed with Cell-FindR v2.0.0 using settings with a quality measure of ≥5 genes. Subsequent analysis was performed with Seurat v3.1.5. Cluster which were not identified by CellFindR due to low cell number, were assigned by validated marker expression. GO-term analysis was performed on genes with $p < 0.01$ for each cell type with DAVID 6.8. Regulon activity was assessed with SCENIC v1.1.2-2. Since SCENIC does not allow batch correction and we identified strong batch effects in the combined dataset, we performed SCENIC analysis independently on the two larger datasets2 and 3 using the cisTarget v8 motif collection mc8nr. Pseudotime analysis was performed with monocle3 v0.2.1. Cell subsets for monocle3 analysis were chosen as indicated. For the analysis of main body follicle cell pseudotime, we chose all main body follicle cell clusters and their derivatives with the exception of differentiated stretch cell and border cell clusters, to allow analysis of cells with epithelial character. To plot gene expression in pseudotime we sorted cells based on their pseudotime value. For germ cell and FSC and pFC pseudotime expression maps we sorted cells into bins of 10 cells each in pseudotime heatmaps. Multicolor UMAP plots were visualized using SCope[81]. Scales in Seurat expression plots and maps display the expression in log((UMI + 1/total UMI)×10^4), monocle3 plot scales display log10(normalized gene expression). Attempts to investigate the RNA velocity exposed too low read numbers for introns in our dataset. This is likely due to the use of polyT primers in 10X datasets and short intron lengths in *Drosophila* genes with little possibilities for binding of polyT primers.

**Fly husbandry.** Flies were reared under standard lab conditions at 25 °C and fed wet yeast for at least three consecutive days prior to dissections. For G-TRACE and RNAi experiments, Gal4-drivers were combined with *tub-Gal80^ts* and bred at 18 °C to repress Gal4 activity during development. For 18 °C G-TRACE controls flies were kept at the restrictive temperature fed wet yeast for at least 3 consecutive days prior to dissection. For 7d and 14d time points adult flies were shifted to 29 °C and fed wet yeast daily until dissection. Starvation experiments were conducted at 29 °C. Flies were kept for 7d and fed wet yeast daily to allow induction of G-TRACE, starved for 24 h in an empty vial with a wet kimwipe, and shifted back to rich diet until dissection. For protein starvation experiments, flies were reared at 18 °C until eclosion and shifted to 29 °C with daily wet yeast for 7d. Flies were then shifted to empty vials containing a kimwipe tissue soaked with 200 mM sucrose solution for 3 days, then fed wet yeast for 4 consecutive days before dissection. For 18 °C controls flies were reared at 18 °C until eclosion and shifted to 29 °C for 14d with daily wet yeast for induction of Gal4 activity before shifting back to 18 °C. We maintained flies at 18 °C with daily wet yeast for 5d to ensure that Gal4 activity was fully abolished, in agreement with established protocols[82], before subjecting flies to starvation in empty vials with a wet kimwipe tissue for 3 days, while control flies were kept on a wet yeast diet continuously. Flies were dissected after a total of 14d post eclosion at 18 °C. For intensity measurements of fax::GFP, control flies were fed wet yeast for three consecutive days, while starved flies were shifted to an empty vial with a wet kimwipe on day 2 and dissected after 24 h starvation.

**Fly stocks.** The following fly stocks were used in this study:

BDSC stocks: *Canton-S* (64349), *Rpn12R-GFP* (36986), *fax::GFP* (50870), *GstS1-lacZ* (11036), *santa-maria-Gal4* (24521), *fax-Gal4* (77520), *Pdk1-Gal4* (76682), *Jupiter::GFP* (6825), *CG46339-Gal4* (77710), *Wnt4-Gal4* (67449), *stl-Gal4* (77732), *pnt::GFP* (42680), *dpp-lacZ* (12379), *cv-2-lacZ* (6342), *Sox14::GFP* (55842), *SPARC::GFP* (56111), *Pvf1-lacZ* (12286), G-TRACE: *UAS-RedStinger, UAS-Flp, Ubi-(FRT.STOP)-Stinger* (28281), *tub-Gal80^ts* (7108), *109-30-Gal4* (7023), *UAS-CD8::RFP* (27399), *UAS-Dp-RNAi* (31767), *UAS-Dref-RNAi* (31941 (Supplementary Fig. 2b) and 35692 (Supplementary Fig. 2c)), *UAS-jim-RNAi* (35609), *UAS-Atf3-RNAi* (26741), *UAS-Myc-RNAi* (36123), *UAS-chinmo-RNAi* (62873), *UAS-foxo* (80564), *UAS-InR^K1409A* (8252), *UAS-Rel*[68] (55778), *UAS-esg-RNAi* (42846), *UAS-Rheb* (9688), *UAS-Tl10B* (58987), *13CO60-Gal4* (47860), *c587-Gal4* (67747), *Wnt4::GFP* (36982), *br^[Z2]::GFP* (38630).

   *tj-Gal4*[83]
   *13CO6-GFP* (generated from BDSC stock 47860)[36]
   VDRC: *Yp1::GFP* (318746), *pnt-RNAi* (7171).
   Kyoto Stock Center (DGRC): *CG9674-Gal4* (112322).
   *hh-lacZ* and *ptc-pelican* (kind gifts from Tom Kornberg), *upd-Gal4* (kind gift from Denise Montell).

**Immunofluorescence staining, imaging, and figure preparation.** Flies were dissected in PBS at RT and ovaries were fixed for 15 min at RT with 4% PFA. Ovaries were washed with PBS twice and blocked for 30 min at RT with blocking solution (PBS with 0.2% Triton X-100 and 0.5% BSA). Primary antibody incubation was performed overnight at 4 °C in blocking solution. On the following day ovaries were washed three times for 10 min with blocking solution and incubated with secondary antibodies diluted in blocking solution for 4 h. After washing with PBS ovaries were mounted in DAPI Fluoromount-G (Thermo Fisher Scientific, OB010020) and imaged using a Zeiss M2 Axioimager with Apotome unit or Nikon C1si Spectral Confocal microscope. Images were analysed with FIJI[84]. The following antibodies were used in this study:

DSHB: mouse anti-gro (anti-Gro, 1:1000), mouse anti-en (4D9, 1:25), mouse anti-Fas3 (7G10, 1:100), rat anti-CadN (DN-EX#8-s, 1:10), mouse anti-Fas2 (1D4, 1:100), mouse anti-BrC (25E9.D7, 1:50), mouse anti-aop (8B12H9, 1:100). rabbit anti-GFP (Cell Signaling #2956, 1:1000), rabbit anti-blanks 1:1000 (kind gift from Erik Sontheimer, 1:1000), rabbit anti-vas (Santa Cruz Biotechnology sc-30210, 1:1000), rabbit anti-cas (kind gift from Ward Odenwald, 1:1000), mouse anti-beta Galactosidase (Promega Z378A, 1:100), chicken anti-beta Galactosidase (abcam ab9361, 1:100), rat anti-RFP (ChromoTek 5F8, 1:1000), guinea pig anti-tj (kind gift from Dorothea Godt, 1:5000), guinea pig anti-zfh1 (kind gift from James Skeath, 1:500), anti-chicken 555 (Sigma-Aldrich SAB4600063, 1:1000). Additional secondary antibodies were purchased from Thermo Fisher Scientific and used at 1:1000: goat anti-rabbit 488 (A-11008), goat anti-rat 555 (A-21434), goat anti-mouse 647 (A-21236), goat anti rabbit 555 (A-21428), goat anti-guinea pig 488 (A-11073), goat anti-guinea pig 555, goat anti-mouse 488 (A-11029), goat anti-mouse 555 (A-21424), goat anti-rat 555 (A-21434).

Images were acquired with either a Zeiss M2 Axioimager with Apotome unit using Axiovision v8.2.0 software or a Nikon C1si Spectral Confocal microscope using EZ-C1 for Nikon C1 Gold Version 3.80 build 860 software. Image processing and analysis was performed with FIJI 2.0.0-rc-69/1.52p[84]. Figures were prepared using Inkscape 1.0. All image raw data can be obtained from the authors upon request.

**Statistics and reproducibility.** All statistical analysis of data and generation of graphs were performed in R. Boxes in box plots show the median and inter-quartile range; lines show the range of values within 1.5× of the interquartile range. Error bars show the S.E.M. The means and S.E.M. values for plots with stacked bars are provided in Supplementary Data 12. All images are representatives of at least two independent experiments and images with associated quantifications (Figs. 3n, o, 5b, d, 7a–d, g–i, Supplementary Fig. 5n and p–q, Supplementary Fig. 9b–c and e–f) are representative of at least three independent experiments.

## Data availability
The authors declare that all data supporting the findings of this study are available within the article and its supplementary information files or from the corresponding author upon reasonable request. The raw data for each figure is provided in a supplementary file (RawData.xlsx). These same data are also in the .RData file (Rust_2020.Rdata), which is in a format that can be accessed by the code in Supplementary software, Supplementary Software 1.Rmd. The raw data for the single-cell sequencing datasets have been deposited in the NCBI GEO database[85] under accession code: GSE136162. Image raw data are available on Mendeley at https://doi.org/10.17632/wtm6sygnmg.3. Other datasets used for comparison of transcriptome profiles are available NCBI GEO database[85] with the accession ID GSE138987[26], GSE4235[69] or from ArrayExpress with the accession number E-MTAB-7063[27]. Source data are provided with this paper.

## Code availability
R scripts to produce plots and perform statistical analysis are available in Supplementary Software file - in Supplementary software 1. Scripts for clustering and filtering in Seurat v3.1.5 is provided in Supplementary Software 2. The monocle3 script is provided in Supplementary Software 3. The SCENIC script is provided in Supplementary Software 4. These data are also available on Mendeley at https://doi.org/10.17632/wtm6sygnmg.3.

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

## Acknowledgements

We thank Tom Kornberg and Denise Montell for sharing fly lines and Dorothea Godt, James Skeath, Erik Sontheimer and Ward Odenwald for antibodies. We are grateful to Marco Conti, Sumitra Tatapudy and Nathaniel P Meyer for critical comments on the manuscript. We also thank the Bloomington Stock Center, the Vienna Drosophila Resource Center, the Kyoto Stock Center (DGRC) and the Developmental Studies Hybridoma Bank for many stocks and resources used in this paper. K.R. is supported by the Deutsche Forschungsgemeinschaft (DFG, project number 419293565), and T.G.N., J.B.S and A.D.T are supported by grants from the National Institutes of Health, GM097158 and GM136348 (T.G.N), DK118421 (J.B.S), and DC018076 (A.D.T).

## Author contributions

K.R. and T.G.N. wrote the manuscript and designed and performed the experiments. K.R., L.E.B., and J.B.S. optimized the protocol for the production of single-cell solutions from *Drosophila* ovaries. K.R. and L.E.B. performed the bioinformatic analysis. K.S.Y. and A.D.T. developed the CellFindR algorithm. K.S.Y. and J.S.P. performed initial CellFindR runs.

## Competing interests

The authors declare no competing interests.
