## [Peer Review File · Nature Communications]

Reviewers' Comments:

Reviewer #2:

Remarks to the Author:

In this revised paper, Rust, Nystul and colleagues describe the use of scRNA-seq to comprehensively characterize cell types within the *Drosophila* germarium. The authors have directly addressed each of the concerns listed in my original critique. In particular, inclusion of a third scRNA-seq dataset and new transient induction of lineage tracing significantly strengthen the overall conclusions. The authors have also added additional experiments that show knockdown of *esg* or over-expression of *Rheb* and *Tl* drives escort cell conversion to follicle stem cells in the absence of starvation. These new data further strengthen the conclusion that specific escort cells can convert to FSCs under the right environmental or genetic conditions. The organization and clarity of the text have been markedly improved, and the addition of new panels and the reorganization of several figures also adds to the strengths of the manuscript. I now fully support the publication of this study.

Reviewer #3:

Remarks to the Author:

Summary

The *Drosophila* ovary is a powerful genetic model for cell biology and physiology. As such, there is great history and continued interest in using the ovary as a model to understand the molecular mechanisms controlling diverse cell biological processes, including stem cell self-renewal, cell fate specification, epithelial morphogenesis, and meiosis. The ovary is a complex organ composed of both germline and somatic cell types at multiple developmental stages. Given the cellular diversity and tissue complexity, transcriptional profiles of individual cell types derived from whole-organ RNA-sequencing have been challenging to interpret. The study by Rust and colleagues seek to overcome this limitation by using single-cell transcriptomic profiling to create a *Drosophila* ovary cell atlas.

Critique

In their revised manuscript, Rust and colleagues present a comprehensive, well-validated cell atlas of the *Drosophila* ovary. The data and analyses presented are novel, timely, and a critical resource needed by researchers in the field. The authors have addressed all of the very extensive comments made by the original three reviewers, creating a much-improved manuscript. The text has been condensed and is a pleasure to read. Importantly, the authors added a third 10x genomics dataset, bringing the total number of cells analyzed to around 14,000 and bolstering the bioinformatic prowess of the study. A strength of the manuscript is that the subpopulations of cells identified in the cell atlas correspond well to cell type identification and lineage relationships previously inferred from genetic mutants and/or antibody/reporter characterization. This demonstrates the utility, robustness, and accuracy of the technique. The authors have provided several new experiments, and collectively these help demonstrate validation of the data. The authors have also made arrangements for the data to be easily accessible by the research community. This is an exciting study, and I have no further suggestions for improving the manuscript.

Reviewer #4:

Remarks to the Author:

The authors have been highly responsive to all comments and concerns raised by Reviewer 1. The analysis of the single-cell data is comprehensive, careful, and well documented. In addition to responding to all analysis-related comments, the authors have added another 10x dataset and have run RNAi experiments to validate TFs and cell transitions.